# Enterovirus D68 seroepidemiology in Taiwan, a cross sectional study from 2017

Jian-Te Lee[1], Wei-Liang Shih[2], Ting-Yu Yen[3], Ai-Ling Cheng[3], Chun-Yi Lu[3], Luan-Yin Chang[3]*, Li-Min Huang[3]

1 Department of Pediatrics, National Taiwan University Hospital, Yun-Lin Branch, Yunlin, Taiwan, 2 Institute of Epidemiology and Preventive Medicine, College of Public Health, National Taiwan University and Infectious Diseases Research and Education Center, Ministry of Health and Welfare and National Taiwan University, Taipei, Taiwan, 3 Department of Pediatrics, National Taiwan University Hospital, College of Medicine, National Taiwan University, Taipei, Taiwan

* lychang@ntu.edu.tw

**Data Availability Statement:** All relevant data are within the paper and its Supporting Information files.

**Funding:** This study was supported by grants from National Taiwan University Hospital, Yun-Lin

## Abstract

### Background

Enterovirus D68 (EV-D68) was discovered in 1962 and has unique characteristics compared to the characteristics of other enteroviruses. There were few documented cases before the epidemic in the United States in 2014. The Taiwan Centers for Diseases Control also confirmed that EV-D68 has been endemic, and some cases of acute flaccid myelitis were reported in Taiwan. To understand the current EV-D68 serostatus, we performed an EV-D68 seroepidemiology study in Taiwan in 2017.

### Methods

After informed consent was obtained, we enrolled preschool children, 6- to 15-year-old students and 16- to 49-year-old people. The participants underwent a questionnaire investigation and blood sampling to measure the EV-D68 neutralization antibody.

### Results

In total, 920 subjects were enrolled from the northern, central, southern and eastern parts of Taiwan with a male-to-female ratio of 1.03. The EV-D68 seropositive rate was 32% (26/82) in infants, 18% (27/153) in 1-year-old children, 43% (36/83) in 2-year-old children, 60% (94/156) in 3- to 5-year-old children, 89% (108/122) in 6- to 11-year-old primary school students, 98% (118/121) in 12- to 15-year-old high school students, 100% (122/122) in 16- to 49-year-old women and 100% (81/81) in 16- to 49-year-old males in 2017. Among preschool children, EV-D68 seropositivity was related to age (p for trend <0.0001), size of household ≧4 members (p = 0.037) and kindergarten attendance (p = 0.027). The seropositive rate varied among different geographic regions.

### Conclusion

EV-D68 infection was prevalent, and its seropositive rates increased with age, larger household size and kindergarten attendance among preschool children.

Branch (grant number NTUHYL107.X016) to J-TL, from the Taiwan Centers for Disease Control, the Ministry of Health and Welfare, Taiwan (grant number MOHW 106-CDC-C-114-000117) to L-YC and the Ministry of Science and Technology, Taiwan (grant numbers MOST 108-2321-B-002-016 and 108-3017-F-002-004) to L-YC. This work was also financially supported by the 'Center of Precision Medicine' from The Featured Areas Research Center Program within the framework of the Higher Education Sprout Project by the Ministry of Education (NTU-108L901401) to L-MH. The funders had no role in the conduct of the study or the preparation of the manuscript.

**Competing interests:** The authors have declared that no competing interests exist.

## Introduction

Enterovirus D68 (EV-D68) was first isolated from four children with pneumonia and bronchiolitis in California in 1962 [1], but it has been reported rarely compared with other enteroviruses. Sporadic cases were mentioned before the epidemic in the United States in 2014, when thousands of cases were reported with an increase in acute flaccid myelitis (AFM) [2,3]. Following this substantial outbreak, EV-D68 was also detected in Canada, Europe, and Asia and subsequently spread worldwide in 2014 [3]. Biennial outbreaks have been recognized in the United States [4] and some European counties [5] since 2014 after the enhancement of AFM surveillance networks and retrospective studies.

The Taiwan Centers for Diseases Control confirmed that EV-D68 circulated in Taiwan, and 92 EV-D68 isolates were identified between 2007 and 2016 [6,7]. In contrast to other enterovirus infections, which usually cause hand, foot and mouth disease or herpangina, EV-D68 tends to cause mild respiratory illness in children but has a propensity to the develop complications, including acute respiratory distress syndrome, especially in those with a preceding history of bronchial asthma [2] and AFM, a rare and devastating condition with no currently available therapy [8]. Therefore, understanding the epidemiology of EV-D68 infection may help to provide sanitary information and develop policies for improving public health. There have been limited data from seroprevalence studies worldwide as well as in Taiwan. To understand the current EV-D68 serostatus, we performed an EV-D68 seroepidemiology study in Taiwan in 2017 and analyzed risk factors associated with EV-D68 seropositivity.

## Materials and Methods

### Study subjects and data collection

The Institutional Review Board of National Taiwan University Hospital approved this study (approved number 201704069RIND). All participants were enrolled in an EV-A71 seroprevalence study in Taiwan in 2017 as described elsewhere [9]. After written informed consent was obtained from parents or guardians of children, we enrolled preschool children, 6- to 11-year-old primary school students and 12- to 15-year-old high school students in the northern (Taipei City), eastern (Hualien County), western (Yunlin County) and southern (Kaohsiung City) regions of Taiwan between May and November 2017. Taipei City and Kaohsiung City are two metropolitan cities, whereas Hualien County and Yunlin County are two rural areas. Adult women and adult men were also enrolled in the four different regions of Taiwan after their written informed consent was obtained.

Participants completed a questionnaire investigation and provided a blood sample, which was submitted for the measurement of EV-D68 neutralization antibody. The questionnaire solicited demographic data, residential area, number of children and adults in a family, sources of drinking water, employment of a babysitter, enrollment in a kindergarten or childcare center, and breastfeeding during infancy. All interviewers were trained, and information was collected from several family members to minimize recall bias. The questionnaires for preschool children, students and adults are listed in S1, S2 and S3 Files.

### Laboratory methods for EV-D68 neutralizing antibody measurement

The neutralizing antibody test for EV-D68 followed the standard protocol of a neutralization test. Serum samples were heat-treated for 30 minutes at 56°C, serially diluted and mixed with 100 50% tissue culture-infective doses (TCID50) of EV-D68, a local circulating strain (GenBank accession number MK371394, genotype B3), and the mixture was incubated for 2 hours at 33°C. Thereafter, rhabdomyosarcoma cells were added to each reaction well and incubated

at 33˚C in a 5% $CO_2$ incubator. Each plate included a cell control, serum control, and virus back-titration. The cytopathic effect was monitored for 5 to 6 days after incubation, and the serotiter was determined when the cytopathic effect was observed in one TCID50 of the virus back-titration. Seropositivity was defined as a serotiter ≥8. For details, please see https://www. protocols.io/view/the-ev-d68-neutralizing-antibody-test-baknicve.

## Statistical analyses

We analyzed the data with SAS statistical software (version, SAS Institute, Cary, North Carolina). We used Student's *t* test for continuous data and chi-square tests for categorical data. Multivariate analysis was performed with multiple logistic regression analysis. The factors with *p*-values < .2 in the univariate analysis were selected for inclusion in the multivariate analysis. A *p*-value < .05 indicated statistical significance.

# Results

## Demography and EV-D68 serostatus in 2017

We conducted nationwide recruitment from urban (northern and southern) and rural (eastern and western) regions, as shown in Table 1. In total, 920 subjects were enrolled from the northern, western, southern and eastern parts of Taiwan, with a male-to-female ratio of 1.03.

The EV-D68 seropositive rate was 32% (26/82) (range: 14–41%) in infants, 18% (27/153) (range: 11–26%) in 1-year-old children, 43% (36/83) (range: 11–61%) in 2-year-old children, 60% (94/156) (range: 48–71%) in 3- to 5-year-old children, 89% (108/122) (range: 81–96%) in 6- to 11-year-old primary school students, 98% (118/121) (range: 94–100%) in 12- to 15-year-old high school students, 100% (122/122) in 16- to 49-year-old women and 100% (81/81) in 16- to 49-year-old men in 2017. The seropositive rate varied among different geographic regions, but the differences were not significantly different in multivariate analysis. Overall, seroprevalence was not related to sex (p = 0.28) after we standardized the rate according to the male to female ratio of the study population, as shown in Fig 1.

School-aged children and adults tended to have higher neutralization antibody titers (≥128) than preschool children: the percentage with higher neutralization antibody titers (≥128) was 11% in children under 3, 37% in 3- to 5-year-old children, 53% in 6- to 15-year-old students and 61% in 16- to 50-year-old people.

**Table 1. Age-specific EV-D68 seropositive rates in different parts of Taiwan in 2017.**

| Age (years) | Total | North | West | South | East | P value |
|---|---|---|---|---|---|---|
| <1 | 32% (26/82) | 41% (12/29) | 40% (10/25) | 14% (3/21) | 14% (1/7) | 0.11 |
| 1 | 18% (27/153) | 26% (8/31) | 19% (6/32) | 18% (8/45) | 11% (5/45) | 0.43 |
| 2 | 43% (36/83) | 61% (17/28) | 39% (9/23) | 11% (1/9) | 39% (9/23) | 0.06 |
| 3–5 | 60% (94/156) | 60% (42/70) | 48% (14/29) | 62% (16/26) | 71% (22/31) | 0.36 |
| 6–11 | 89%(108/122) | 87% (26/30) | 96% (26/27) | 91% (31/34) | 81% (25/31) | 0.28 |
| 12–15 | 98% (118/121) | 100% (31/31) | 97% (30/31) | 100% (28/28) | 94% (29/31) | 0.31 |
| Women (16–49) | 100% (122/122) | 100% (30/30) | 100% (30/30) | 100% (31/31) | 100% (31/31) | NA |
| Men (16–49) | 100% (81/81) | 100% (21/21) | 100% (20/20) | 10% (20/20) | 100% (20/20) | NA |

The P value was measured by the chi-square test. The North (Taipei City) and South (Kaohsiung City) regions are metropolitan areas, whereas the East (Hualien County) and West (Yunlin County) regions are rural areas. Numbers in parentheses are the numbers of participants with EV-D68 seropositivity/the number of participants tested.

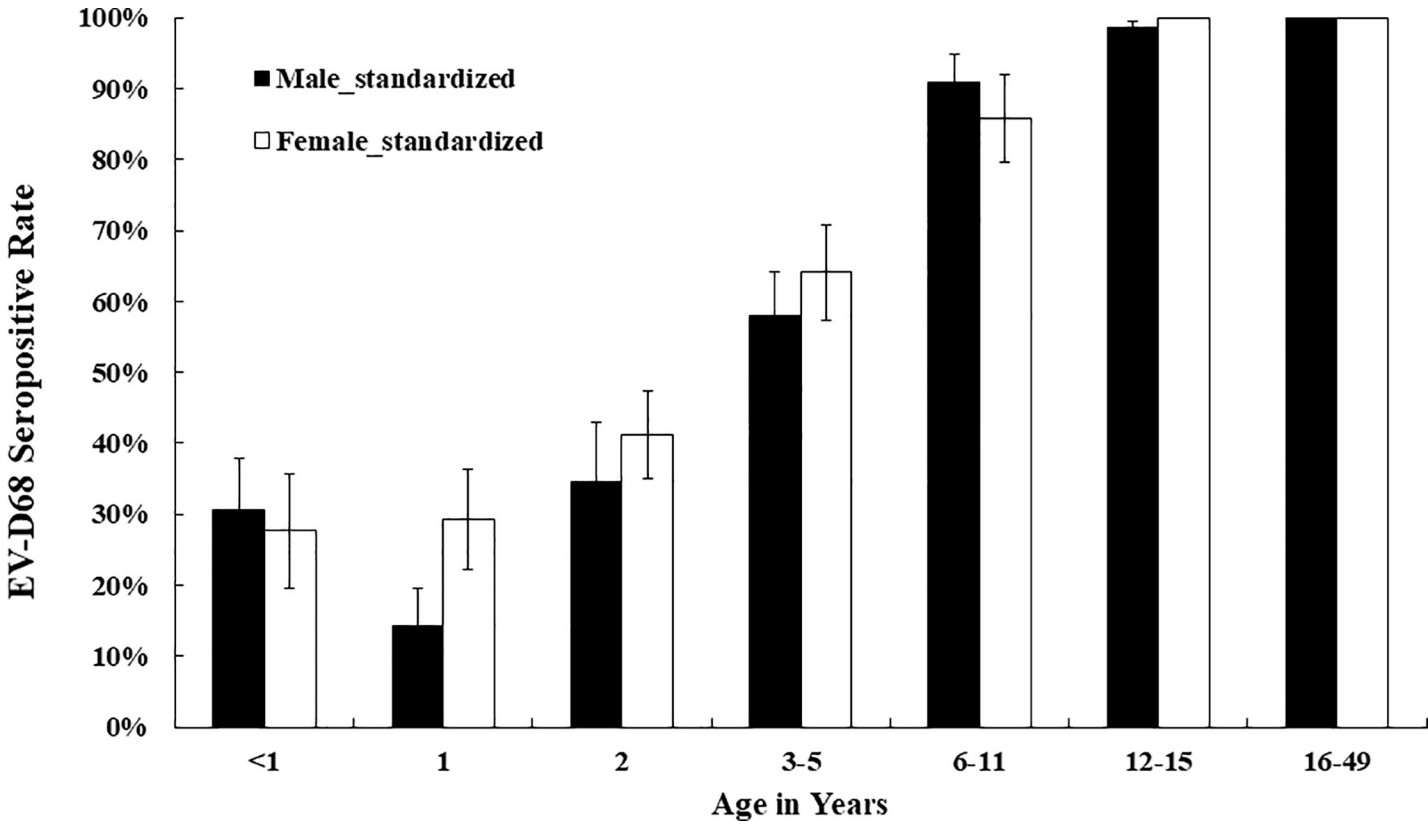

**Fig 1. Age-specific EV-D68 serostatus between males and females in 2017.** The bars demonstrate the mean seropositive rate ± standard error.

### Risk factors associated with EV-D68 seropositivity in preschool and school-aged children

We performed univariate and multiple logistic regression analyses to define the risk factors among preschool children (Table 2). We identified age, region, size of households, siblings and kindergarten/daycare attendance as the significant factors in the univariate analysis. There was a significant correlation between household size and the number of siblings (Spearman correlation, $r_s$ = 0.36, p-value<0.0001), so we selected household size for multivariate analysis.

Seropositive infants under 1 year of age were considered to have maternally transferred antibodies and were therefore omitted. Among 1- to 5-year-old preschool children, EV-D68 seropositivity was related to age (p for trend <0.0001), size of household ≥4 members (p = 0.037) and kindergarten attendance (p = 0.027) in multivariate analysis. Compared with 1-year-old children, 2-year-old children were 3-times more likely (odds ratio [OR], 3.42; 95% confidence interval [CI], 1.83–6.41; p = 0.0001) and 3- to 5-year-old children were 4-times more likely (OR, 4.55; 95% CI, 2.47–8.38; p<0.0001) to be seropositive. Preschool children with a household size ≥4 members had a significantly higher seropositive rate (OR, 1.35; 95% CI, 1.03–2.64; p = 0.037) than children whose household size was ≤3 members. Preschool children attending kindergarten/daycare had a significantly higher seropositive rate (OR, 2.18; 95% CI, 1.09–4.35; p = 0.027) than those not attending kindergarten/daycare.

The EV-D68 seropositive rate of school-aged children increased with age. The risk of 12- to 15-year-old children being seropositive was higher (OR, 5.10; 95% CI, 1.43–18.23; p = 0.01) than that of 6- to 11-year-old children.

**Table 2. Risk factors associated with EV-D68 seropositivity in preschool children younger than 6 years of age in 2017.**

| Factor | | $\chi^2$/Wald[*] | df | P-value | OR | 95% CI |
|---|---|---|---|---|---|---|
| **Univariate analysis** | | | | | | |
| Sex | | 0.79 | 1 | 0.374 | | |
| Age | | 59.56 | 2 | <0.0001 | | |
| Region | | 11.80 | 3 | 0.008 | | |
| Size of household, # | | 7.25 | 1 | 0.007 | | |
| Siblings, # | | 17.26 | 1 | <0.0001 | | |
| Kindergarten/Daycare attendance | | 25.91 | 1 | <0.0001 | | |
| **Multivariate analysis with dummy variables** | | | | | | |
| Age | 1 | | | | 1 | |
| | 2 | 14.78 | 1 | 0.0001 | 3.42 | (1.83, 6.41) |
| | 3–5 | 23.59 | 1 | <0.0001 | 4.55 | (2.47, 8.38) |
| Region | North | | | | 1 | |
| | West | 0.66 | 1 | 0.417 | 0.76 | (0.40, 1.47) |
| | South | 2.28 | 1 | 0.132 | 0.61 | (0.32, 1.16) |
| | East | 0.01 | 1 | 0.905 | 0.96 | (0.51, 1.81) |
| Size of household | ≤3 | | | | 1 | |
| | ≥4 | 4.37 | 1 | 0.037 | 1.65 | (1.03, 2.64) |
| Kindergarten/Daycare attendance | No | | | | 1 | |
| | Yes | 4.86 | 1 | 0.027 | 2.18 | (1.09, 4.35) |

OR = odds ratio; CI = confidence interval

[*] the values for univariate and multivariate analysis were $\chi^2$ and Wald values, respectively.

#There was a significant correlation between household size and the number of siblings (Spearman correlation, $r_s$ = 0.36, *p*-value<0.0001), so we selected household size for multivariate analysis.

## Discussion

In this study, the seroprevalence of EV-D68 in Taiwan in 2017 increased with age, from 43% by two years of age to nearly 100% in individuals 12 years of age and older. Before the active surveillance of EV-D68 in cases with acute flaccid paralysis by the Taiwan Centers for Disease Control (CDC) beginning in July 2015, EV-D68 may have been prevalent and circulating in Taiwan. Our results were in line with studies from China [10], the United States [11], the United Kingdom [12] and the Netherlands [13].

Genotyping of EV-D68 isolates revealed that different subtypes co-circulated in Taiwan. Subclade B3 was the major circulating genotype after 2014 [7] and was used for the testing of neutralizing antibodies in our study. EV-D68 seroprevalence could have been higher if all co-circulating genotypes were tested.

High EV-D68 seroprevalence was noted in the United States before the outbreak in 2014 [11], raising the question about seropositivity and seroprotection. It has also been proposed that neutralizing antibodies could result from infections by other enteroviruses [14]. Our earlier study, however, showed seroconversion in confirmed EV-D68 infected children, all of whom developed mild respiratory symptoms [15]. A recent study demonstrated significantly higher antibodies to EV peptides in cerebrospinal fluid (CSF) of patients with AFM than controls [16]. Among AFM patients, 43% (6/14) of CSF samples and 74% (8/11) of sera were immunoreactive to an EV-D68-specific peptide, which was in contrast to the non-immunoreactivity in either CSF or sera from the controls [16]. A Japanese study also confirmed that serum neutralization antibody titers against EV-D68 increased during outbreaks but waned

**Table 3. Comparison of the age-specific EV-D68 serostatus among different countries.**

| | Country, Year | | | | | | | | | |
|---|---|---|---|---|---|---|---|---|---|---|
| | Taiwan | China | Kansas City, Missouri, USA | | UK | UK | The Netherlands | | The Netherlands | |
| Age | 2017 | 2010 [18] | 2012–2013 [11] | | 2006 [12] | 2016 [12] | 2006–2007 [13] | | 2015–2016 [13] | |
| Strain* | B3 | Synthetic* | Fermon, B1# | B2, A2 | B3 | B3 | Fermon | B3 | Fermon | B3 |
| <1 | 32% (26/82) | 79% (96/121)a 20% (33/109)b | NA | | 75% (9/12)c 0% (0/6)d | 69% (31/45)c 43% (9/21)d | 94% (17/18) | 44% (8/18) | 95% (19/20) | 68% (13/19) |
| 1 | 18% (27/153) | NA | NA | | 55% (44/80)e | 72% (62/86)e | 82% (18/22)h | 82% (18/22)h | 95% (19/20)h | 90% (18/20)h |
| 2 | 43% (36/83) | 44% (34/77) | 100% | 60%, 81% | | | | | | |
| 3–5 | 60% (94/156) | | | | | | | | | |
| 6–11 | 89% (108/122) | 83% (65/78) | 100% | 83%, 89% | 63% (31/49)f | 90% (46/51)f | | | | |
| 12–19 | ~100% | | 100% | 93%, 98% | 81% (92/113)g | 95% (90/95)g | 85% (17/20)i | 95% (19/20)i | 95% (19/20)i | 100% (20/20)i |
| 20–29 | | NA | | | ~90% | ~95% | 100% (20/20)j | 95% (19/20)j | 100% (20/20)j | 100% (20/20)j |

The cutoff for EV-D68 seropositivity was ≥1:8 [12,13,18], except in the UK study, which used a titer of >1:16 as the cutoff [12]. *The name or genotype of the viral strain was used for neutralizing antibody, and reverse genetics with Fermon strain was used to produce the EV-D68 virus (Synthetic) in the China study. #The results in the USA were the same as those for either the Fermon or B1 strain.

NA: not available. Numbers in parentheses are the number of individuals with EV-D68 seropositivity/the number of individuals tested.

a The rate for infants aged 1 to 5 months and

b the rate for infants aged 6 months to 1 year [18].

c The rate for infants aged under 6 months

d the rate for infants aged 6 months to 1 year

e the rate for 1- to 4-year-old children

f the rate for 5- to 9-year-old children and

g the rate for 10- to 19-year-old students [12].

h The rate for 1- to 10-year-old children

i the rate for 11- to 20-year-old children and

j the rate for 21- to 30-year-old adults [13].

over one year without outbreaks [17]. Since the disease spectrum and pathogenesis of EV-D68 are not fully understood, most studies have focused on severe cases, such as AFM and/or acute respiratory distress syndrome. A comprehensive and prospective study is needed in the future to better understand the associated disease burden. Nevertheless, our seroprevalence study and others provide estimations of the disease burden.

The EV-D68 seroprevalence rate of 1- to 5-year-old children was approximately 50% in 2006 and 75% in 2016 in the United Kingdom [12]. The EV-D68 seroprevalence rate was approximately 59% among children younger than 15 years old in China, and it was positively correlated with age among 1-year-old (10%) to 15-year-old (92%) children [18]. The seroprevalence rates of adults approach 100% in China, the United States, the United Kingdom, the Netherlands and Taiwan [10–13,18]. We thus listed and compared EV-D68 serostatus among different countries in Table 3. All these seroprevalence studies are comparable, and the EV-D68 seropositive rates do not vary greatly among different countries. The above finding could imply that EV-D68 has spread extensively worldwide, although only a limited number of severe cases have been reported.

The risk factors associated with EV-D68 seroprevalence among preschool children in our study included age, larger household size and daycare/kindergarten attendance. An earlier study performed in a kindergarten in Taiwan from 2006 to 2008 revealed 9 cases of EV-D68 confirmed by viral isolation in the autumn of 2007, and the EV-D68 seroprevalence of children aged between two and five years increased from 19% (25/130) at baseline in 2006 to 67% (83/124) at the end of the study in 2008 [15]. The seroconversion rate of 49 children with initial seronegative and paired sera was 73% (36/49), which indicates that preschool children are highly susceptible to EV-D68 infection and that the transmission rate within kindergartens/daycares is very high. This current study also highlights that the risk of EV-D68 infection is twofold higher if preschool children attend daycare or kindergarten. Children with a larger household size had a significantly higher seropositive rate, implying that a larger household size may be associated with a higher risk for EV-D68 household transmission.

Moreover, we found a very low (4–8%) EV-A71 seropositive rate among preschool children in the same study population [9]. The Taiwan CDC has established surveillance networks and adopted infection control measures against several EV-A71 outbreaks since 1998. The low seroprevalence of EV-A71 in young children partly reflects successful containment by public health policies. In contrast, the high seroprevalence of EV-D68 in the same population pinpointed the completely different clinical manifestations of EV-D68 even though the route of transmission of both viruses may be similar. Unlike EV-A71 infections associated with typical hand, foot and mouth disease, non-specific mild upper respiratory symptoms in most EV-D68 cases will not warn teachers, caregivers, families, clinicians or health authorities to take strict preventive measures against transmission. Given the different clinical manifestations from other enteroviruses, preventive measures against EV-D68 should be reconsidered in terms of prospective surveillance and education. Since most EV-D68 infections are asymptomatic or only cause mild symptoms, the disease burden might be underestimated via respiratory specimen culture or PCR examination. Consequently, seroepidemiological data in this and other studies, as shown in Table 3, provide more accurate information on the spread of this infection.

There are some limitations in this study. First, the test population characteristics (sampling methods, geographical and demographical characteristics) were not the same among different countries, although the laboratory method was the same. Second, the EV-D68 viral strains or genotypes used for neutralizing antibodies are different among different countries, as shown in Table 3. Although the seroprevalence rates were not comparable, we tried to clarify the differences and performed some important comparisons among countries.

## Conclusions

We found fairly high EV-D68 seropositive rates in children younger than 15 years old and that rate reached 100% in adults. Age, larger household size and kindergarten/daycare attendance are the most significant risk factors associated with EV-D68 seropositivity among preschool children.

## Supporting information

**S1 File. Questionnaire for preschool children with Chinese-English parallel texts.**
(PDF)

**S2 File. Questionnaire for students with Chinese-English parallel texts.**
(PDF)

**S3 File. Questionnaire for adults with Chinese-English parallel texts.**
(PDF)

## Author Contributions

**Conceptualization:** Jian-Te Lee, Luan-Yin Chang.

**Data curation:** Jian-Te Lee, Ting-Yu Yen.

**Formal analysis:** Wei-Liang Shih, Luan-Yin Chang.

**Funding acquisition:** Jian-Te Lee, Luan-Yin Chang, Li-Min Huang.

**Methodology:** Wei-Liang Shih, Ai-Ling Cheng, Luan-Yin Chang.

**Resources:** Ting-Yu Yen, Luan-Yin Chang, Li-Min Huang.

**Supervision:** Luan-Yin Chang, Li-Min Huang.

**Validation:** Chun-Yi Lu.

**Writing – original draft:** Jian-Te Lee, Luan-Yin Chang.

**Writing – review & editing:** Wei-Liang Shih, Ting-Yu Yen, Ai-Ling Cheng, Chun-Yi Lu, Luan-Yin Chang, Li-Min Huang.

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
