## [Decision Letter · Decision Letter 0]

23 Jan 2020

PONE-D-19-34975

Enterovirus D68 seroepidemiology in Taiwan, a cross sectional study in 2017

PLOS ONE

Dear Prof. Chang,

Your manuscript has now been reviewed by two experts in the field. One of them also reviewed your previous paper on the same topic. While bother reviewers are supportive of publicaion, they have also raised substantive concerns about the presentation and technical quality of your work. Particularly, it was felt that the prresentation should be substantially improved by seeking the help from a professional or native writer. In addition, one reviewer suggested reanalysis of some data (e.g. Table 2) using another method. 

After careful consideration, we feel that it has merit but does not fully meet PLOS ONE’s publication criteria as it currently stands. Therefore, we invite you to submit a revised version of the manuscript that addresses the points raised during the review process.

We would appreciate receiving your revised manuscript by Mar 08 2020 11:59PM. To enhance the reproducibility of your results, we recommend that if applicable you deposit your laboratory protocols in protocols.io, where a protocol can be assigned its own identifier (DOI) such that it can be cited independently in the future. For instructions see: http://journals.plos.org/plosone/s/submission-guidelines#loc-laboratory-protocols

We look forward to receiving your revised manuscript.

Kind regards,

Dong-Yan Jin

Academic Editor

PLOS ONE

Additional Editor Comments:

Please address the concerns raised satisfactorily as they might re-review your revised manuscript.

Reviewers' comments:

Reviewer's Responses to Questions

**Comments to the Author**

1. Is the manuscript technically sound, and do the data support the conclusions?

Reviewer #1: No

Reviewer #2: Yes

2. Has the statistical analysis been performed appropriately and rigorously? 

Reviewer #1: No

Reviewer #2: Yes

3. Have the authors made all data underlying the findings in their manuscript fully available?

Reviewer #1: Yes

Reviewer #2: Yes

4. Is the manuscript presented in an intelligible fashion and written in standard English?

Reviewer #1: Yes

Reviewer #2: No

5. Review Comments to the Author

Reviewer #1: The logic of data analysis is wrong, hence, the results maybe not right.

To analyse the risk factor for the infection, firstly, the univariate analysis maybe applied, if the P<0.2, the factor is entered into multivariate analysis, if P>=0.2, the factor can not be entered into multivariate analysis. Only in multivariate analysis, the dummy variable be needed and meanly meaningful. Hence, please reanalyze data (for instance, table 2).

The reference maybe useful for you. ”Risk factors for Blastocystis infection in HIV/AIDS patients with highly active antiretroviral therapy in Southwest China”.

Reviewer #2: The study by Lee et al describes a serosurvey of EV-D68 neutralising antibodies among the Taiwanese population as well as trying to identify factors associated with seropositivity. The study was well-designed and the cohort size was substantial. Results demonstrate a remarkably high overall seropositivity in an age-dependent manner similar to results of other studies.

Comments:

Importantly, the manuscript should be proof read by a native English speaker as there are many grammatical or inaccuracies throughout the manuscript. This would improve the readability and clarity of the manuscript considerably.

- Table 1: in the age category, please add ages to the adult women/men category.

- Table 3: The study in the Netherlands investigated neutralizing Abs against two EV-D68 strains, please indicate to which data (Fermon strain or B3 clinical) is referred to in this table. It would be helpful to indicate for each study, which D68 strain was used.

- The data from table 3 could also be presented in a chart allowing for an easier comparison of the seropositivity

- Fig. 1: I would suggest bar graphs instead of a connected line. Since the average for all investigated areas is given, please show also the mean +- SD.

6. PLOS authors have the option to publish the peer review history of their article (what does this mean?). If published, this will include your full peer review and any attached files.

Reviewer #1: No

Reviewer #2: No

---

## [Author Response · Author response to Decision Letter 0]

23 Feb 2020

Point-by-point response

Dong-Yan Jin

Academic Editor

PLOS ONE

To enhance the reproducibility of your results, we recommend that if applicable you deposit your laboratory protocols in protocols.io, where a protocol can be assigned its own identifier (DOI) such that it can be cited independently in the future. For instructions see: http://journals.plos.org/plosone/s/submission-guidelines#loc-laboratory-protocols

Response:

We have deposited our laboratory protocols on the website protocols.io. For details, please see https://www.protocols.io/view/the-ev-d68-neutralizing-antibody-test-baknicve.

Reviewer #1: 

The logic of data analysis is wrong, hence, the results maybe not right. To analyse the risk factor for the infection, firstly, the univariate analysis maybe applied, if the P<0.2, the factor is entered into multivariate analysis, if P>=0.2, the factor cannot be entered into multivariate analysis. Only in multivariate analysis, the dummy variable be needed and meanly meaningful. Hence, please reanalyze data (for instance, table 2).

The reference may be useful for you. ”Risk factors for Blastocystis infection in HIV/AIDS patients with highly active antiretroviral therapy in Southwest China”.

Response:

We used the analytical method you suggested. The factors with p-values <.2 in the univariate analysis were selected for multivariate analysis. In our study, we identified age, region, size of household, number of siblings and kindergarten/daycare attendance as significant factors in the univariate analysis. There was a significant correlation between household size and the number of siblings (Spearman correlation, rs=0.36, p-value<0.0001), so we selected household size for multivariate analysis. In the multivariate analysis, we identified age, household size, and kindergarten/daycare attendance as risk factors for EV-D68 seropositivity. Table 2 was modified, as shown in the revised manuscript. We also added some discussion regarding household size: Children with a larger household size had a significantly higher seropositive rate, implying that a larger household size may be associated with a higher risk for EV-D68 household transmission. We hope the reanalysis better demonstrates the characteristics of our study. 

Reviewer #2: The study by Lee et al describes a serosurvey of EV-D68 neutralising antibodies among the Taiwanese population as well as trying to identify factors associated with seropositivity. The study was well-designed and the cohort size was substantial. Results demonstrate a remarkably high overall seropositivity in an age-dependent manner similar to results of other studies.

Importantly, the manuscript should be proof read by a native English speaker as there are many grammatical or inaccuracies throughout the manuscript. This would improve the readability and clarity of the manuscript considerably.

Response:

Thank you very much for your comments. The manuscript has been carefully proofread by a native English speaker. The editing certificate was in the supplementary materials.

- Table 1: in the age category, please add ages to the adult women/men category.

Response: We have added ages to the adult women/men category in Table 1

- Table 3: The study in the Netherlands investigated neutralizing Abs against two EV-D68 strains, please indicate to which data (Fermon strain or B3 clinical) is referred to in this table. It would be helpful to indicate for each study, which D68 strain was used.

Response: We added the strain used in the Dutch study and other studies.

- The data from table 3 could also be presented in a chart allowing for an easier comparison of the seropositivity

Response: Thank you very much for your suggestion. We would like to make a chart but could not complete it due to different age groups, different viral strains and too many countries in all the studies. Therefore, we are sorry to keep it as Table.

- Fig. 1: I would suggest bar graphs instead of a connected line. Since the average for all investigated areas is given, please show also the mean +- SD.

Response: According to your suggestion, we switched to bar graphs including mean +- SE, as shown in the revised manuscript.

---

## [Editor Report · Decision Letter 1]

25 Feb 2020

Enterovirus D68 seroepidemiology in Taiwan, a cross sectional study in 2017

PONE-D-19-34975R1

Dear Dr. Chang,

Thank you for submitting your revised manuscript.

I have read your response and your paper carefully. I am convinced that you have satisfactorily addressed all concerns raised by the two reviewers. I am also sure that your paper merits publication although there is an earlier study of yours  on this topic. Documentation of the new dataset is fully justified. 

Therefore, we are pleased to inform you that your manuscript has been judged scientifically suitable for publication and will be formally accepted for publication once it complies with all outstanding technical requirements.

With kind regards,

Dong-Yan Jin

Academic Editor

PLOS ONE

Additional Editor Comments:

Well done!
---

## [Editor Report · Acceptance letter]

28 Feb 2020

PONE-D-19-34975R1 

Enterovirus D68 seroepidemiology in Taiwan, a cross sectional study from 2017 

Dear Dr. Chang:

I am pleased to inform you that your manuscript has been deemed suitable for publication in PLOS ONE. Congratulations! Your manuscript is now with our production department. 

With kind regards,

on behalf of

Professor Dong-Yan Jin 

Academic Editor

PLOS ONE